# PeerJ

# Physical activity and pre-diabetes—an unacknowledged mid-life crisis: findings from NHANES 2003–2006

Kathryn Farni[1], David A. Shoham[2], Guichan Cao[2], Amy H. Luke[2], Jennifer Layden[2], Richard S. Cooper[2] and Lara R. Dugas[2]

[1] Stritch School of Medicine, Loyola University Chicago, Maywood, IL, USA
[2] Department of Public Health Sciences, Stritch School of Medicine, Loyola University Chicago, Maywood, IL, USA

Corresponding author
Lara R. Dugas, ldugas@lumc.edu

## ABSTRACT

The prevalence of pre-diabetes (PD) among US adults has increased substantially over the past two decades. By current estimates, over 34% of US adults fall in the PD category, 84% of whom meet the American Diabetes Association's criteria for impaired fasting glucose (IFG). Low physical activity (PA) and/or sedentary behavior are key drivers of hyperglycemia. We examined the relationship between PD and objectively measured PA in NHANES 2003–2006 of 20,470 individuals, including 7,501 individuals between 20 and 65 yrs.We excluded all participants without IFG measures or adequate accelerometry data (final $N = 1,317$). Participants were identified as PD if FPG was 100–125 mg/dL (5.6–6.9 mmol/L). Moderate and vigorous PA in minutes/day individuals were summed to create the exposure variable "moderate-vigorous PA" (MVPA). The analysis sample included 884 normoglycemic persons and 433 with PD. There were significantly fewer PD subjects in the middle (30.3%) and highest (24.6%) tertiles of PA compared to the lowest tertile (35.5%). After adjusting for BMI, participants were 0.77 times as likely to be PD if they were in the highest tertile compared to the lowest PA tertile ($p < 0.001$). However, these results were no longer significant when age and BMI were held constant. Univariate analysis revealed that physical activity was associated with decreased fasting glucose of 0.5 mg/dL per minute of MVPA, but multivariate analysis adjusting for age and BMI was not significant. Overall, our data suggest a negative association between measures of PA and the prevalence of PD in middle-aged US adults independent of adiposity, but with significant confounding influence from measures of BMI and age.

## BACKGROUND

The prevalence of pre-diabetes among US adults has increased markedly over the past two decades (*Cowie et al., 2006*; *Karve & Hayward, 2010*). By current estimates, over 34% of nondiabetic US adults can be classified as pre-diabetic, 84% of whom meet the American Diabetes Association's criteria for impaired fasting glucose (IFG) (*American Diabetes Association, 2008*). Individuals with IFG and/or impaired glucose tolerance (IGT) are at increased risk for developing diabetes and cardiovascular disease (CVD)

compared to those with normal fasting plasma glucose levels. In addition, IFG has been shown to be an independent predictor of CVD mortality after adjustment for age, sex, and other traditional CVD risk factors (*Deedwania & Fonseca, 2005*; *Barr et al., 2007*). Compared with pre-diabetes, type 2 diabetes is associated with even greater risk for adverse cardiovascular health outcomes. Therefore, preventing the progression from pre-diabetes to diabetes is a critical link in our efforts to improve cardiovascular outcomes regardless of impact on other cardiovascular risk factors (*Knowler et al., 2002*).

Investigations into the relationship between physical activity and insulin levels unequivocally demonstrate that high levels of sedentary time, low levels of daily movement, and little moderate to vigorous physical activity are associated with poor glycemic control (*Helmerhorst et al., 2009*; *Mayer-Davis et al., 1998*; *Colberg, 2012*; *Assah et al., 2008*). Insulin resistance is a fundamental attribute of both IFG and IGT, and the inverse relationship between physical activity and insulin resistance is amply documented in both healthy individuals and those with pre-diabetes (*Dela et al., 1992*; *Dube et al., 2011*; *Hawley & Lessard, 2008*). Additionally, there is strong evidence that a dose–response relationship exists between insulin sensitivity and exercise "dose" (a combination of intensity, duration, and frequency) (*Dube et al., 2012*). Individuals with pre-diabetes have been shown to benefit significantly from intensive lifestyle intervention programs that include modified diets, increased physical activity, or a combination of the two (*Gillies et al., 2007*; *Gregg et al., 2012*). In 2002, a randomized clinical trial (The Diabetes Prevention Program [DPP]) demonstrated that lifestyle intervention reduced the incidence of Type 2 diabetes by 58% among those at high risk for developing the disease, including participants deemed to be pre-diabetic (*Knowler et al., 2002*). However, while intensive lifestyle intervention may prevent the progression from pre-diabetes to diabetes, once a patient is diagnosed with type 2 diabetes, interventions such as diet, exercise, and weight loss do not appear to be effective in reducing cardiovascular morbidity and mortality (*Look AHEAD Research Group, 2013*). Additionally, the most effective behavioral intervention techniques aimed at increasing physical activity among Type 2 diabetics have yet to be elucidated (*Kinmonth et al., 2008*). Nonetheless, early lifestyle intervention in those with pre-diabetes may represent a window of opportunity for health improvement before the irreversible effects of diabetes set in.

Recently, the American Diabetes Association updated their pre-diabetes screening guidelines to reflect the growing body of research that demonstrates early intervention, especially among those with other underlying cardiovascular risks, may significantly decrease the morbidity and mortality related to glucose intolerance (*Ferrannini, 2014*; *American Diabetes Association, 2014*). All asymptomatic individuals with BMI $\geq$ 25 and at least one risk factor for the development of diabetes (including physical inactivity) should undergo screening using A1C, FPG, or oral glucose tolerance test, and those without risk factors should undergo screening beginning at age 45. The American Diabetes Association has long recommended that individuals with IFG and/or IGT undergo lifestyle modifications that include a 5–10% weight loss and moderate intensity physical activity for 30 min/day (*Nathan et al., 2007*). To date, studies on the prevalence of IFG among US

adults have rarely included objective data on measures of physical activity. Unfortunately, self-reporting of behaviors such as physical activity are subject to both random and systematic reporting bias and are notoriously unreliable (*Duncan et al., 2001*). Devices that objectively measure movement intensity provide a reliable alternative to self-reported activity. To better estimate the physical activity patterns in individuals with IFG, we examined NHANES 2003–2006 to assess the relationship between IFG and physical activity using accelerometers.

## METHODS

Data from NHANES 2003–2006 were used in this analysis. NHANES samples a representative portion of the US civilian non-institutionalized population via a complex, multistage probability design. Study participants underwent a two-hour interview at home, followed two weeks later by a clinic examination at a mobile examination center (MEC). All participants provided informed consent at at-home interviews and MEC examinations. NHANES annually samples fifteen geographic locations throughout the US, and the overall data collection process for each location takes several weeks. Individual clinical examinations last 3–4 h and all data collection methods are standardized to minimize site-specific bias (*Centers For Disease Control And Prevention, 2014*). The NHANES sample from 2003–2006 included a total of 20,470 individuals, including 6,932 non-pregnant individuals between the ages of 20–65. We excluded all participants with a history of heart disease ($n = 446$), diabetes ($n = 183$), chronic bronchitis, emphysema, and renal disease ($n = 183$). Additionally, patients without valid BMI, fasting glucose, ethnic, and accelerometry data were excluded ($n = 4,803$) for a final sample size of $n = 1,317$.

Standard measuring procedures were used to record height (m), weight (kg), and body mass index (BMI, kg/m$^2$) (*National Center For Health Statistics, 2005*). Fasting plasma glucose (FPG) levels were determined by analyzing blood samples taken from individuals who had fasted for the previous 8–24 h. Participants were identified as pre-diabetic if FPG was 100–125 mg/dL (5.6–6.9 mmol/L) (*American Diabetes Association, 2008*).

To monitor physical activity, participants were asked to wear an Actigraph (Actigraph, LLC, Ft. Walton Beach, FL) model 7164 accelerometer over their right hip (*National Center For Health Statistics, 2005*). Accelerometers measure and record vertical acceleration "counts" as indicators of the wearer's physical activity (*Welk, Schaben & Morrow, 2004*). Participants were asked to wear the device while they were awake in order to achieve a minimum of 10 h of monitor wear time on four or more days. Wear time was determined by subtracting non-wear time from 24 h, where non-wear time was defined as an interval of 60 or more consecutive minutes with zero activity intensity counts. The 2003–2006 NHANES survey participants wore the accelerometer for 7 consecutive days.

Physical activity as measured by the accelerometers was recorded according definitions of moderate and vigorous physical activity put forth by *Freedson, Melanson & Sirard (1998)* and described in *Troiano et al. (2008)*. These definitions use weighted averages to allow the measurements of vertical acceleration recorded by the accelerometers to be translated into estimates of actual physical activity intensity. To meet the requirements for

moderate activity, the accelerometers must record 2,020 counts/min, while a classification of vigorous activity requires the accelerometers to record at least 5,999 counts/min. These values were used to estimate the total number of minutes per day the individual spent at each activity level. Physical activity data is presented here as the number of 1 and 10 min sustained "bouts" of activity at a given level, where bouts are defined as 1 or $\geq 10$ consecutive minutes above the relevant threshold, allowing for interruptions of 1 or 2 min below threshold. The number of bouts is then averaged over the total number of days during which the accelerometer was worn. Following the guidelines described by Troiano, we allowed for up to 3 min of below threshold count activity before considering the bout to be ended (*Troiano et al., 2008*). Mean counts per minute were calculated by dividing the sum of activity counts for a day by the number of minutes of wear-time in that day across all valid days.

All statistical analyses were completed using Stata (Stata Corp, College Station, TX).

## RESULTS

We analyzed 884 healthy control subjects and 433 subjects with pre-diabetes for measures of moderate to vigorous activity in one- and ten-minute bouts as measured by accelerometer. To ensure that data were representative of the US population, we applied mobile exam clinic (MEC) survey weights to NHANES data using standard Stata survey commands. Each group was analyzed for differences in age, BMI, and levels of moderate to vigorous physical activity. The pre-diabetic group was significantly older ($p < 0.001$) and had a larger BMI ($p < 0.001$) than the healthy controls. Male subjects had significantly higher FPG levels than female subjects ($p < 0.001$). Subjects were grouped into tertiles by their average number of bouts of moderate to vigorous physical activity, with the first tertile accumulating the least amount of MVPA (Table 1). There were significantly lower proportions of pre-diabetic subjects within the second (30.3%) and third (24.6%) tertiles as compared to the first (35.5%) ($p < 0.001$), and at least 50% of the prediabetics were found in the lowest tertile. Male subjects were disproportionately distributed among the tertiles of physical activity, with the lowest, middle, and highest tertiles containing 33.5%, 53.2%, and 74.0% males, respectively ($p < 0.001$). When controlling for BMI alone, subjects within the third tertile were 0.77 times as likely to be pre-diabetic than those within the first tertile ($p < 0.01$). However, when holding BMI, sex, race, poverty income ratio, and age constant, the difference in odds between the first and third tertile was no longer significant ($p > 0.01$). Using regression analysis and controlling for sex, we explored the associations between fasting glucose, age, and BMI, all as continuous variables. Univariate analysis revealed that physical activity was associated with decreased fasting glucose such that for every minute of additional MVPA, fasting blood glucose levels fell by 0.15 mg/dL ($p < 0.001$). Additionally, multivariate analysis of physical activity and fasting blood glucose adjusting for age, sex, and BMI revealed that physical activity was associated with a decreased fasting blood glucose of 0.04 mg/dL for every minute of additional MVPA ($p = 0.001$). Lastly, we identified those individuals who met the United States Surgeon General's recommended daily physical activity ($>$30 min of

**Table 1** Subject characteristics (mean ± SD) of reweighted analyzed sample—NHANES 2003–2006, ages 20–65 y for all subjects ($N = 1,317$) by tertile of moderate to vigorous physical activity (tertile 1 $N = 463$, tertile 2 $N = 422$, tertile 3 $N = 432$). Thresholds for tertile 1 = 0–16 one minute bouts, 0–12.7 ten minute bouts; Tertile 2 = 16.4–33.9 one minute bouts, 0–27 ten minute bouts; Tertile 3 = >33.9 one minute bouts, 0–125 ten minute bouts.

| Variable | Tertile 1 ($N = 463$) | Tertile 2 ($N = 422$) | Tertile 3 ($N = 432$) | All subjects ($N = 1,317$) |
|---|---|---|---|---|
| Weight (kg) | 85.3 ± 1.1 | 82.7 ± 1.1 | 79.8 ± 0.8 | 82.6 ± 0.6 |
| Age (yr) | 45.0 ± 0.6 | 42.0 ± 0.7 | 39.1 ± 0.6 | 42.1 ± 0.4 |
| Percent male | 33.5% | 53.2% | 74.0% | 52.7% |
| BMI | 30.1 ± 0.3 | 27.7 ± 0.3 | 26.5 ± 0.2 | 28.1 ± 0.2 |
| SBP (mmHg) | 122.1 ± 0.8 | 118.6 ± 0.9 | 116.9 ± 0.6 | 119.2 ± 0.5 |
| DBP (mmHg) | 72.4 ± 0.8 | 72.2 ± 0.7 | 70.9 ± 0.6 | 71.9 ± 0.4 |
| Insulin (uu/mL) | 11.8 ± 0.4 | 9.6 ± 0.4 | 7.9 ± 0.3 | 9.8 ± 0.2 |
| Glucose (mg/dL) | 96.7 ± 0.6 | 95.7 ± 0.6 | 94.3 ± 0.6 | 95.6 ± 0.4 |
| Mean counts per minute | 244.9 ± 3.8 | 343.3 ± 4.7 | 501.4 ± 7.0 | 361.4 ± 5.5 |
| Mean wear time (hr/day) | 14.1 ± 0.1 | 14.5 ± 0.1 | 14.7 ± 0.1 | 14.4 ± 0.0 |
| Moderate activity (min. in 1 min bouts) | 8.7 ± 0.2 | 24.2 ± 0.3 | 51.9 ± 1.2 | 28.0 ± 0.8 |
| Vigorous activity (min. in 1 min bouts) | 0.1 ± 0.0 | 0.7 ± 0.1 | 3.1 ± 0.3 | 1.3 ± 0.1 |
| Moderate and vigorous activity (min. in 1 min bouts) | 8.8 ± 0.2 | 24.9 ± 0.3 | 55.0 ± 1.4 | 29.3 ± 0.8 |
| Moderate activity (min. in 10 min bouts) | 0.7 ± 0.1 | 5.2 ± 0.4 | 14.5 ± 0.8 | 6.7 ± 0.4 |
| Vigorous activity (min. in 10 min bouts) | 0.0 ± 0.0 | 0.5 ± 0.1 | 2.0 ± 0.2 | 0.8 ± 0.1 |
| Moderate and vigorous activity (min. in 10 min bouts) | 0.8 ± 0.1 | 6.0 ± 0.4 | 18.2 ± 1.1 | 8.2 ± 0.5 |
| Pre-diabetic (%) | 35.5 ± 3.1 | 30.3 ± 3.2 | 24.6 ± 2.6 | 30.2 ± 2.3 |

moderate to vigorous physical activity per day). Before adjusting for age and BMI, fasting blood glucose was 2.1 mg/dL lower among those who met the surgeon general's physical activity guidelines when compared to those who do not meet the guidelines ($p = 0.001$). However, when controlling for age and BMI, these between group differences were no longer statistically significant.

## DISCUSSION

Our study was the first of its kind to use a large national database to examine the relationship between objectively measured physical activity and pre-diabetes. A significant strength of the current study was the use of objective measures of physical activity. However, variations in participant compliance with accelerometry wear time remain a potential source of sampling bias. Nonetheless, the close association between pre-diabetes and physical activity confirms that across the limited range of physical activity engaged in by US adults, diabetes can at least be postponed by relatively little extra activity. However, age, sex, and BMI remained significant confounding factors. We found that NHANES subjects who were the most physically active were 0.77 times as likely to be pre-diabetic as their BMI matched controls who were not as physically active, but these effects were erased when controlling for age. Even among those participants who achieved the recommended 30 min of daily moderate to vigorous physical activity, a decrease in IFG could not be attributed to physical activity alone.

Previous studies have described high rates of physical function limitations among type 2 diabetics (*Kalyani et al., 2010*; *Gregg et al., 2000*), but the relationship between physical disability and prediabetes has not been studied as intensively. In one of the only investigations of physical function in pre-diabetic adults, *Lee et al. (2013)* found that a high prevalence of physical function limitations among those with pre-diabetes. Taken together, these results suggest that counseling and other interventions focused on increasing physical activity among those at risk for development of type 2 diabetes should be targeted on young adults when musculo-skeletal impairments are less likely to make physical activity more of a challenge.

In NHANES, mean IFG crosses 100—the threshold for pre-diabetes—at age 40. This mid-life period of asymptomatic mild hyperglycemia therefore represents a crucial window of opportunity for intervention to forestall diabetes and its associated cardiovascular and metabolic morbidities. Additionally, the inseparable relationship between age and physical inactivity necessitates early intervention not only on the grounds of improving blood glucose profiles, but is also in preventing or delaying the onset of physical disability. Although a causative relationship cannot be established from this cross-sectional data, our findings are consistent with the intervention literature regarding physical activity and glucose control. While routine screening for diabetes remains controversial, the gradual age-related climb in IFG is an important personal health parameter. Persons in this age range should receive more attention from primary care physicians, be encouraged to value their good health and take seriously the opportunities available to them to prevent or postpone the onset of frank diabetes.

### Funding
There was no funding for this study.

### Competing Interests
Richard S. Cooper is an Academic Editor for PeerJ. None of the other authors have any competing interests to declare.

### Author Contributions
- Kathryn Farni and Lara R. Dugas conceived and designed the experiments, analyzed the data, wrote the paper, prepared figures and/or tables, reviewed drafts of the paper.
- David A. Shoham analyzed the data, wrote the paper, reviewed drafts of the paper.
- Guichan Cao analyzed the data, wrote the paper, prepared figures and/or tables, reviewed drafts of the paper.
- Amy H. Luke and Jennifer Layden wrote the paper, reviewed drafts of the paper.
- Richard S. Cooper conceived and designed the experiments, analyzed the data, wrote the paper, reviewed drafts of the paper.

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
