# Peer review of "Physical activity and pre-diabetes—an unacknowledged mid-life crisis: findings from NHANES 2003–2006"

_PeerJ, doi:10.7717/peerj.499_

## Round 0.1 · original submission · Major Revisions

The reviewers feel considerable extra work is required. Please address these on a point by point basis and in particular justify the exclusion of the Cambridge Physical activity intervention study.

Reviewer 1 ·

Basic reporting

See below

Experimental design

See below

Validity of the findings

See below

Additional comments

This is a very short cross-sectional study that demonstrates that low physical activity coincides with pre-diabetes.

From the data in this format, it is not possible to assign any causality, and thus, the final sentence “A successful population-level intervention to prevent or delay type II diabetes must raise the PA within this age cohort” is an overstatement based on these data.

The authors have failed to include the Cambridge physical activity intervention study in the Lancet in 2008, showing little effect from an intensive PA intervention.

The Look Ahead trial is also informative in this regard.

The authors undertook a regression model here, but only briefly report it. A full table report may be more useful.

Overall the results are preliminary and speculative.

Reviewer 2 ·

Basic reporting

"No Comments"

Experimental design

Methods:
1. It is unclear why the sample was limited to 20-65 y, and would benefit from a rationale. On Line 69, the age range is also indicated as 20-45 y.

2. The analytic sample is N=1,317. For comparative purposes, i) a full description of how many participants were excluded at each stage (from the initial sample of 20,470), and ii) a missing value analysis for important socio-demographic and clinical characteristics would be instructive.

3. Analysis: It is unclear why additional sociodemographic characteristics were not included in Table 1, or why only BMI and age were adjusted for. Level of statistical significance is not included in the table. For logistic regression, was regression adjusted matching used? (line 133).

4. Table 1: The title indicates that the analysis uses a "reweigthed sample". Details of any rescaling of survey weights should be included in the methodology.

Validity of the findings

The use of NHANES (with objectively assessed physical activity) and clinical correlates of pre-diabetes is a major strength. Study purpose and conclusions are appropriately aligned.

Additional comments

While the study purpose is clear, the analyses are under-developed and lack transparency in decisions around refinement of the final sample, choice of confounders, and model building procedures. Specifically, it is unclear why further multivariable analyses (including a broader profile of potential confounders) was not completed, and would have strengthened the analysis considerably. Were any interactions probed? If sex- and ethnic-based analyses were not conducted due to power constraints, this detail would be beneficial.

---

## Round 0.2 · Minor Revisions

Your revisions have thrown up further issues with one of the reviewers. Please respond addressing these quesions fully

Reviewer 2 ·

Basic reporting

No comments

Experimental design

No comments

Validity of the findings

Thank you for the opportunity to review a revision of the manuscript, and for the authors’ responses and points of clarification. Following are several questions regarding presentation of the results that remain.

Major Comments:

1. Analysis: I may have missed it, but I do not see any rationale for adjusting for only BMI, age, and (in some places?) sex. There may be a good reason, but from the text, it is not clear why a broader profile of confounders was not considered, along with potential interactions. Restrictions on number of figures or tables notwithstanding, logistic regression findings might be more clearly presented in a separate table (rather than text only), and would allow for various models to be compared. [See first review].

2. Sample Weights: Were data weighted to be representative of the U.S. population? I do not see any additional information on rescaled weights (Stata procedures) that the authors are referring to in the revised manuscript. [See first review].

3. Table 1. The rational for not presenting p-values in the table is not clear, as the authors have highlighted statistical significance multiple times in the results section (e.g. lines 138-151).

4. Surprisingly, sex is not included in Table 1 as a descriptive. I suspect that a rationale for sex-specific tertiles of activity could also be made, given the population-level differences in PA prevalence in NHANES.

Minor Comments / Discretionary Edits:

5. That Lee et al. (2013) found a high prevalence of physical function limitations in those with PD would speak to the issue of reverse causality; however, in the following sentence, the authors do indeed suggest PA interventions towards this population (lines 173-7).

6. What is the approximate cut-off for PA tertiles (for average number of bouts of MVPA)? This is necessary to contextualize this work (e.g. lines 161-3). Please add to Table 1 legend.

7. Why was only 2003-2006 data used, given that more recent NHANES cycles are available?

8. What differential bias might be present from study exclusions (regarding accelerometer wear time, etc.)?

Additional comments

Outside of the methodological questions noted above, this is an interesting and well-written paper that has clear implications for public health (and the primary prevention of type 2 diabetes).

---

## Round 0.3 · Minor Revisions

You still need to respond to existing concerns of the reviewer

Reviewer 2 ·

Basic reporting

Background and context of the article are excellent.

Minor points:
i) Evidence that the working sample is representative of the U.S. population is not provided. I question whether applying MEC weights to a subsample without recalibration will indeed yield a representative weighting. (i.e. many exclusions are applied, and no missing value analysis is done to compare the original and analytic samples).

ii) Table 1 should be stand-alone and would benefit from inclusion of accelerometer (tertile) thresholds in the legend.

Experimental design

No Comments

Validity of the findings

No Comments

Additional comments

No Comments

---

## Round 0.4 · accepted · Accept

Thankyou for resubmitting a revised version

Your changes are acceptable